# Comparative transcriptome provides molecular insight into defense-associated mechanisms against spider mite in resistant and susceptible common bean cultivars

Abdul Hadi Hoseinzadeh[1][ᵒ]*, Aboozar Soorni[2][ᵒ]*, Marie Shoorooei[1], Masoud Torkzadeh Mahani[3], Reza Maali Amiri[1], Hossein Allahyari[4], Rahmat Mohammadi[1]

**1** Department of Agronomy and Plant Breeding, Faculty of Agriculture, University of Tehran, Karaj, Iran,
**2** Department of Biotechnology, College of Agriculture, Isfahan University of Technology, Isfahan, Iran,
**3** Department of Biotechnology, Institute of Science, High Technology and Environmental Science, Graduate University of Advanced Technology, Kerman, Iran, **4** Department of Plant Protection, Faculty of Agriculture, University of Tehran, Karaj, Iran

ᵒ These authors contributed equally to this work.
* ahzadeh@ut.ac.ir (AHH); soorni@cc.iut.ac.ir (AS)

## Abstract

Common bean (*Phaseolus vulgaris* L.) is a major source of proteins and one of the most important edible foods for more than three hundred million people in the world. The common bean plants are frequently attacked by spider mite (*Tetranychus urticae* Koch), leading to a significant decrease in plant growth and economic performance. The use of resistant cultivars and the identification of the genes involved in plant-mite resistance are practical solutions to this problem. Hence, a comprehensive study of the molecular interactions between resistant and susceptible common bean cultivars and spider mite can shed light into the understanding of mechanisms and biological pathways of resistance. In this study, one resistant (*Naz*) and one susceptible (*Akhtar*) cultivars were selected for a transcriptome comparison at different time points (0, 1 and 5 days) after spider mite feeding. The comparison of cultivars in different time points revealed several key genes, which showed a change increase in transcript abundance via spider mite infestation. These included genes involved in flavonoid biosynthesis process; a conserved *MYB-bHLH-WD40* (MBW) regulatory complex; transcription factors (TFs) TT2, TT8, TCP, Cys2/His2-type and C2H2-type zinc finger proteins; the ethylene response factors (ERFs) ERF1 and ERF9; genes related to metabolism of auxin and jasmonic acid (JA); pathogenesis-related (PR) proteins and heat shock proteins.

## Introduction

Common bean (*Phaseolus vulgaris* L.), is one of the most important edible foods in the world which provides about 50% of the grain legumes for direct human consumption [1–3]. In addition, it is an inexpensive healthy food due to having the richest sources of proteins (20–25%),

**Data Availability Statement:** All RNA-Seq data were deposited to the NCBI sequence read archive (SRA) under the project PRJNA482175.

**Funding:** The authors received no specific funding for this work.

**Competing interests:** The authors have declared that no competing interests exist.

micronutrients and calories [2]. Common bean is widely distributed around the world. In Asia, most collections exist in India [4], and Iran [5]. Notably, over the last 10 years, the production of common bean has increased ~33% in Asia [2]. Because of high nutrient content and commercial potential, common bean holds great promise for fighting hunger and increasing income. Low yield of this crop is attributed to pest attack, weak soil fertility, drought and salinity, and poor agronomic practices [6]. According to the records, the two spotted spider mite (TSSM), *Tetranychus urticae* Koch, is the most widespread and the most polyphagous herbivores mites which feed on cell contents of common bean and causes serious substantial economic losses (up to 100% yield losses) in fields and greenhouses [7, 8]. The TSSM damages plant cells by its stylet that pierces the leaf either in between epidermal pavement cells or through a stomatal opening, suck-out their contents and forms the chlorotic lesions at the feeding sites [8–10]. In recent years, it has become evident that insect-resistant crops have brought great benefits, not only in terms of economic, but also because of the reduction of pesticides use and keeping a safe environment. The development of new cultivars is being established as one of the most appropriate methods and the main objective of plant breeding programs for resistance to TSSM [11, 12]. However, lacking information on how plant and mite interact with each other emphasizes the importance of a comprehensive study of the molecular interactions between common bean and *T. urticae* to understand the mechanisms and potential biological pathways of common bean resistance. Although RNA-Seq has been used to study the expression profiles of stress response genes in model and non-model plants, but there has not been any study of common bean transcriptome changes due to spider mite feeding. In this study we used RNA-Seq analysis to detect differences in gene expression between two cultivars of *P. vulgaris* (susceptible and resistant), and specify effective genes and pathways in response to *T. urticae* infestation. Such information could lead to identity resistant mechanisms and genes in common bean and improve the breeding efforts by identifying molecular markers to incorporate resistance into commercial bean varieties.

## Materials and methods

### Plants and insect infestation

According to our previous study [13], two cultivars, including *Akhtar* and *Naz* were selected as susceptible and resistant cultivars to *T. urticae*, respectively. Seeds were sown in plastic pots (15 cm diameter, 25 cm high) containing soil, peat moss, and perlite (1:1:1), with only a single plant in each pot. The experiment was conducted using a factorial experiment based on completely randomized design with three replicates in greenhouse condition (28 ± 3$^{\circ}$C temperature, 40–50% relative humidity, photoperiod 16h light and 8h darkness). The founder population of mite was collected from a commercial bean farm in Karaj, Iran and colonies were reared on the similar cultivars to be tested for three generations before being used in this experiment. In six-leaf stage (about 30 days after planting) based on Meier [14], 45 same-aged adult female mites were placed on sixth leaf of plants. Since, the number of females is higher than males and they also cause the most damages, only female mites were selected from a mass which had the opportunity for mating. The leaves were collected after 0, 1 and 5 days of infestation. Treated leaves were frozen in liquid nitrogen and kept at −80˚C until they were used for RNA extraction.

### RNA isolation and transcriptome sequencing

Total RNA was extracted from two biological replicates, which each were pooled samples from at least three plants using TRIzol$^{\circledR}$ Reagent (Invitrogen) as described by the manufacturer's protocol, and then treated with RNase-free DNaseI (Invitrogen). Nanodrop™ 2000 spectrophotometer,

agarose gel electrophoresis and Bioanalyzer 2100 (Agilent) were used to check and confirm quantity and quality of RNAs. All RNAs were sent to Beijing Genomic Institute (BGI) in China for library preparation and transcriptome sequencing using the Illumina HiSeq 2500 platform to generate Paired-end (2×150 bp) reads.

### Reads preprocessing and differentially expression analysis

The raw reads were downloaded from BGI institute web site in Fastq format and then were subjected to quality control (QC) analysis using Trimmomatic software [15] to trim low quality reads, adapters and other Illumina-specific sequences, minimum length 50 bp and minimum quality 30 determined as quality thresholds. Before and after filtering, the quality of the raw sequences was assessed with FastQC (https://www.bioinformatics.babraham.ac.uk/projects/fastqc/). Clean reads were mapped to the *P. vulgaris* [16] and *T.urticae* [17] reference genomes V. 2.0 (https://phytozome.jgi.doe.gov) via RNA-Seq aligner STAR software [18] requiring at least 90% of the read sequence to match with at least 95% identity. The STAR-resultant. *bam* files were used to estimate the abundance of mapped reads, differential expression analyses, and visualization of analyses results using Cufflinks [19] package coupled with CummeRbund [20]. Cufflinks was used to calculate FPKM values and differential expression analysis was done with Cuffdiff. The analysis focused on genes with statistically significant difference in expression levels between times and cultivars. The genes were considered significantly differentially expressed if false discovery rate (FDR, the adjusted P value) was <0.01 and Log2 FPKM (fold change) was ≥1.0. All RNA-Seq data were deposited in the NCBI SRA database under the project PRJNA482175.

### GO terms and KEGG pathways enrichment analysis

The functions of the DEGs were characterized using AgriGO's Singular Enrichment Analysis (SEA) module to identify the enriched Gene Ontology terms (http://bioinfo.cau.edu.cn/agriGO/analysis.php) with the agriGO database [21]. The enrichment analysis was performed at significance level of 0.05. Enrichment analysis of Kyoto Encyclopedia of Genes and Genomics (KEGG) pathways was also carried out on DEGs using KOBAS 3.0 web server (http://kobas.cbi.pku.edu.cn/). KEGG pathways with corrected p-value ≤0.05 were considered as significantly enriched.

### Quantitative real-time PCR validation

To validate candidate differentially expressed genes (DEGs), qRT-PCR was performed for six DEGs and *Actin 11* as a reference gene with three replicates. Primers were designed by Primer 3.0 [22] (Table 1), and cDNAs were synthesized by using TaKaRa cDNA Synthesis Kit (TaKaRa, Dalian, China) according to the manufacturer's instructions. The 20 μL qRT-PCR solutions contained EvaGreen Master Mix (Solis Biodyne, 5x), 0.3 μL forward and 0.3 μL reverse primers, and 30 ng of cDNA template. qRT-PCR reactions (95 ˚C, 3 min; 95 ˚C, 5 s; 60 ˚C, 34 s; 40 cycles) were carried out on a Bio-Rad iQ5 Optical System (Bio-Rad Laboratories, CA, USA). Finally, relative gene expression was calculated using $2^{-\Delta\Delta CT}$ formula and REST software [23].

## Results and discussion

### Quality control and mapping statistics

A total of 12 RNA libraries were sequenced with the number of reads ranging from 26.8–30.2 million paired-end reads (Table 2). Approximately 70–73% of reads passed the quality control

**Table 1. Description of the candidate gens and primer sequences for qRT-PCR assay efficiencies.**

| Funcional annotation | Gene ID | Forward primer | Reverse prime |
|---|---|---|---|
| Pathogenesis-related protein | Phvul.004G155500 | TGGGATACAGCTACAGCATCGT | ATCTTCATTGGGTGGAGCATCT |
| WRKY transcription factor 50 | Phvul.009G080000 | GTCGCTGAGATCGGAGAATC | GCAAATCCAGCTTTGACCAT |
| Heat shock protein (Molecular chaperone) | Phvul.008G011400 | CTTTCAACACCAACGCCATG | GCTCAAGCTCCGAGTAGG |
| Leucine Rich Repeat | Phvul.008G044600 | CTTGACTATGAGCTTGTCCCC | TGCTTTCTCTGTAAGGTGTCC |
| MYB113 | Phvul.008G038200 | GTCGCTGAGATCGGAGAATC | GCAAATCCAGCTTTGACCAT |
| Xyloglucan endotransglucosylase/hydrolases | Phvul.005G111300 | AGTTCGACGAGCTGTTCCAG | ACGTTGGTCTGCACGCTGTA |

and an 84.58–90.30% of the clean reads were mapped to unique location in the common bean reference genome. Alignment of clean reads to *T. urticae* reference genome was also carried out to determine whether a significant mite RNA contamination exists in our datasets. Assessment of quality of mRNA-Seq data revealed less than 0.1% mapping, indicating a strong enrichment of genes specific for *P. vulgaris* in all samples.

## Differentially expressed genes (DEGs)

Differentially expressed genes analysis was performed for the pairwise comparisons of twelve libraries. The largest differences in expression occurred among three time points of resistant cultivar. When comparing the different time points for resistant cultivar, 274 differentially expressed genes were identified (S1 Table), almost the same number of DEGs for susceptible cultivar (270 DEGs, S2 Table). The number of up-regulated genes was higher than down-regulated genes in all different time point comparisons in both cultivars. To gain a better understanding, the overlap differentially expressed patterns of DEGs were analyzed between cultivars in each time point and across time points in each cultivars using Venn diagram. The comparison of cultivars in each time points revealed 48, 65 and 81 up-regulated genes along with 46, 59, 68 down-regulated genes in resistant cultivar for control samples and these infested at 1 and 5 days post-feeding, respectively (Fig 1, S3–S5 Tables). The number of DEGs showed a rising trend with the extension of infestation time, So that the smallest and largest differences were observed between resistant and susceptible plants at first and third time points, in which 94 and 146 DEGs were identified, respectively. This result indicates there are probably no significant differences in gene expression patterns during the first attempts of spider mite in both susceptible and resistant reactions. However, gene expression patterns were

**Table 2. Summary of sample information and transcriptome sequencing output statistics for the RNA-Seq libraries.**

| Cultivars | Replicate | Time points | Reads before quality control | Reads after quality control | Removing percent |
|---|---|---|---|---|---|
| *Akhtar* (susceptible) | Replicate 1 | Control | 30261354 | 21020563 | 30.54 |
| | | 1 Day | 28834008 | 20871380 | 27.62 |
| | | 5 Day | 26832585 | 19131482 | 28.70 |
| | Replicate 2 | Control | 31261254 | 22020354 | 29.56 |
| | | 1 Day | 27836208 | 20971682 | 24.66 |
| | | 5 Day | 26632382 | 19231180 | 27.79 |
| *Naz* (resistant) | Replicate 1 | Control | 30593227 | 22305624 | 27.09 |
| | | 1 Day | 27677349 | 19152514 | 30.80 |
| | | 5 Day | 29214322 | 21461892 | 26.54 |
| | Replicate 2 | Control | 30243118 | 22212456 | 26.55 |
| | | 1 Day | 28123454 | 19252314 | 31.54 |
| | | 5 Day | 28876322 | 21863542 | 24.29 |

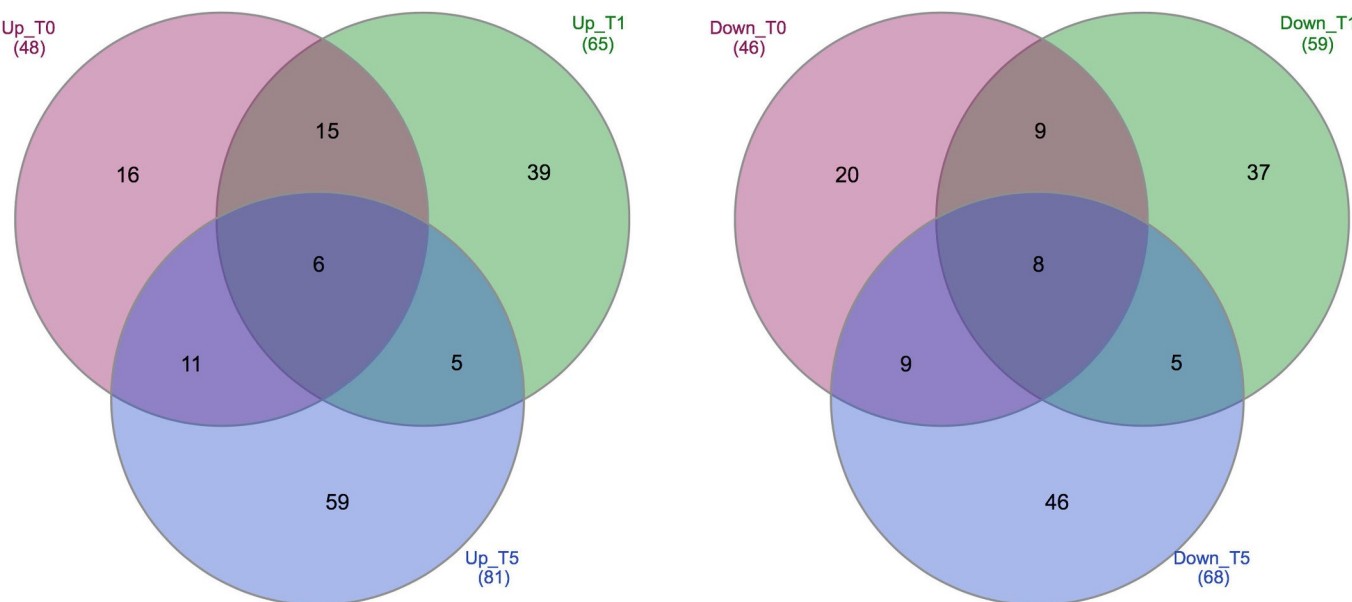

**Fig 1. Venn diagram showing the number of specific and shared DEGs between pair time points in both cultivars.** Up_T0, genes up-regulated in resistant cultivar in comparison with susceptible cultivar at first time point. Down_T0, genes down-regulated in resistant cultivar or up-regulated in susceptible cultivar at first time point. T1 and T3 represent the second and third time points, respectively.

more different during the second phase of infestation depending on the resistance/susceptibility of the plant.

Gene expression patterns were also different with the extension of infestation time depending on the resistance/susceptibility of common bean cultivars. Among DEGs, approximately 44% and 37% of up-regulated genes were common among three time points, while less than 7% of down-regulated genes were shared among times in both cultivars. Interestingly, there was no any common up or down regulated gene between T0T1 (comparison of first and second time points) and T1T3 (comparison of second and third time points) and also any unique gene for T0T3 (comparison of first and third time points). As shown in the Venn diagram in Fig 2, the number of up-regulated DEGs in T0T3 comparison was higher than T0T1 comparison in both cultivars. This analysis indicated that more than 70% and 85% of DEGs were common between T0T1 and T0T3 comparisons, respectively.

## Functional classification and GO enrichment analysis of DEGs

To eliminate the effects of genetic differences, DEGs were compared between cultivars at the same time points. GO analysis detected 57, 39 and 46 categories, according to biological process (P), molecular function (F), and cellular component (C) among all up- and down-regulated genes at time 0, 1 and 3, respectively.

## Secondary metabolism

The differences between control samples was determined by the metabolism of phenylpropanoid, a central to produce defense-related compounds [24–26], including anthocyanins and flavonoids (Fig 3). The two most induced genes involved in flavonoid biosynthesis process corresponded to *dihydroflavonol reductase* (*DFR*) and *chalcone synthase* (*CHS*). We found transcripts of *DFR* which were up-regulated in resistant cultivar (Fig 4). *DFR* has been previously reported as induced by *Fusarium oxysporum* inoculations on *Linum usitatissimum* [27].

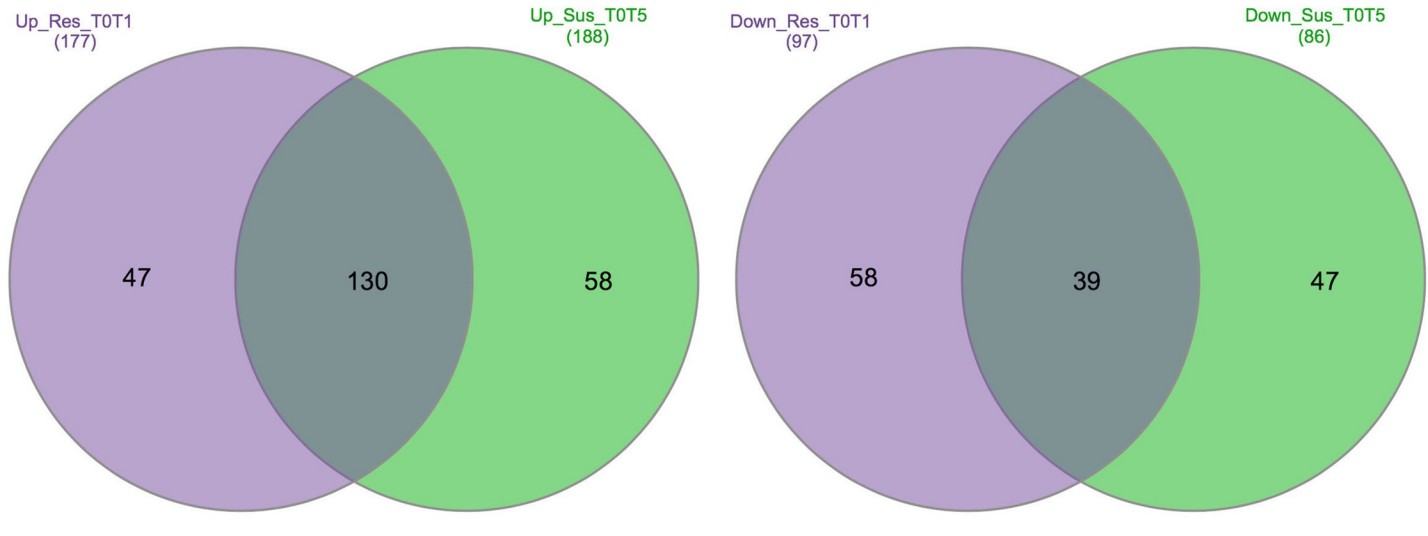

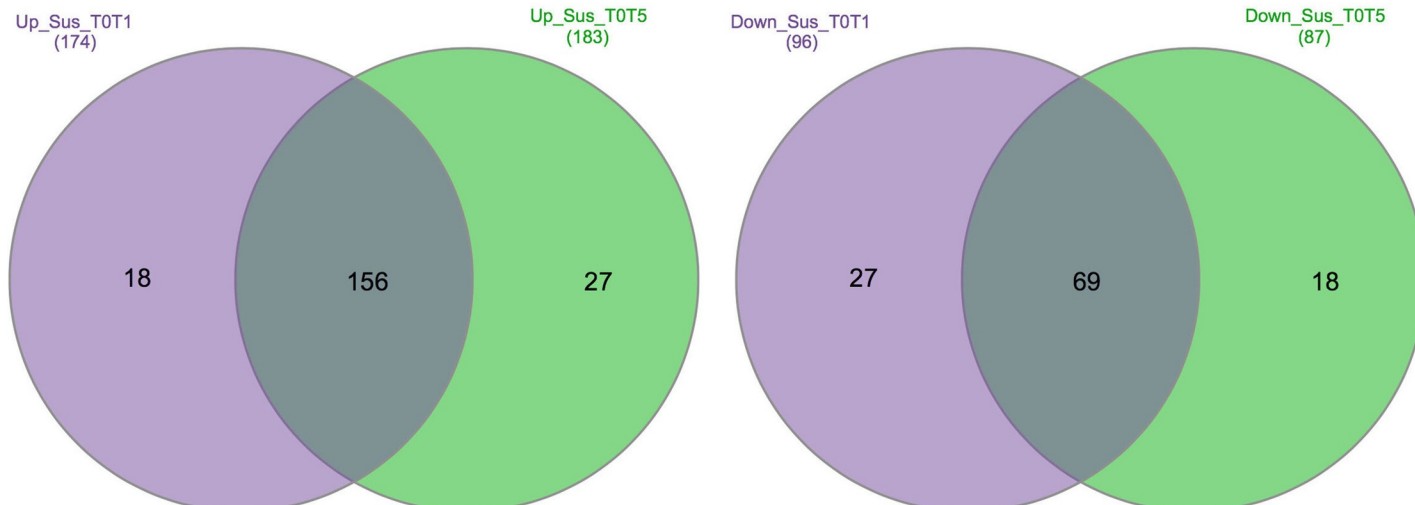

**Fig 2. Venn diagram representing specific and shared up- and down-regulated genes between two sets in each cultivar.** Pink circle (first set) represents the comparison of first and second time points and green (second set) is the comparison of first and third time points. Up and Down represent up-regulated, down-regulated genes. Res and Sus represent resistant and susceptible cultivars.

*Colletotrichum camelliae* on *Camellia sinensis* [28] and *Elsinoe ampelina* in grapevine [29]. *DFR* is also a key regulatory gene belonging to the subgroup of late anthocyanin biosynthesis genes which can be activated by TFs such as MYBs [30, 31]. *CHS* showed more than 7-fold change increase in resistant cultivar. *O-methyltransferases* (*OMTs*), involved in phenylpropanoids, flavonoids, and anthocyanin methylation, was up-regulated in resistant cultivar and showed an 8-fold increase in expression level upon spider mite feeding. *O-Methylation* plays key roles in plant defense following pathogen attack [32]. Additionally, the interaction network suggests the possible involvement of Cytochrome P450 (*CYP*) genes (*CYP72A7* and *CYP71A26*) in resistance, previously reported in several studies [33, 34]. In the susceptible

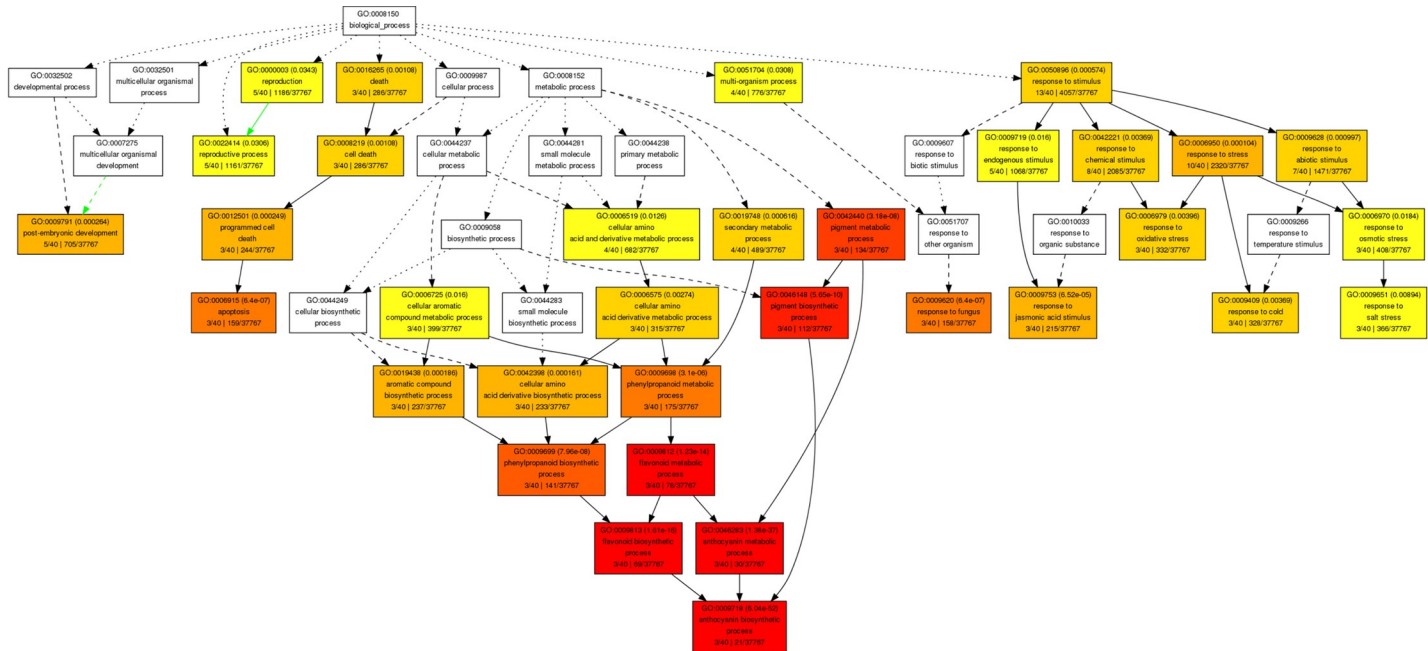

**Fig 3. GO enrichment analysis for up-regulated genes in resistant cultivar as compared to susceptible cultivar at day 0 (control samples).** Boxes in the graph represent GO IDs, term definitions and statistical information. Significant GO terms (p ≤ 0.05) are marked with color. The degree of color saturation of a box is positively correlated to the enrichment level of the term.

genotype, *CYP83B1* gene required for the synthesis of indole glucosinolates, was down-regulated during infestation which is not consistent with previous reports in a number of researches [27, 35]. This result demonstrates the role of spider mite effectors in suppressing the defence-related gene of common bean in compatible interaction (susceptible cultivar) but not in an incompatible interaction (resistant cultivar). Previous investigation indicate when a pathogen interacts with plant tissues, more intensive transcriptional changes are found in the compatible interaction and pathogens commonly develope effectors that interfere with signaling pathways to suppress resistance responses. Finally, other secondary metabolism gene (BAS: beta-amyrin synthase) with higher transcript abundance in resistant cultivar was related to sesquiterpenoid and triterpenoid biosynthesis.

## Transcriptional regulation

TF families found in our study are widely reported to be involved in plant defense responses, including MYB, WRKY, ethylene responsive factors (ERFs), zinc finger domain proteins and basic helix-loop-helix (bHLH). A key TF that currently appears in the studies of plant-pathogen interactions [27, 36] and had a significant expression in resistant cultivar of our study is MYB113. *TRANSPARENT TESTA4* (*TT4*), a chalcone and stilbene synthase family protein, is a key enzyme involved in the biosynthesis of flavonoids to encode *chalcone synthase* (*CHS*), and is required for the accumulation of purple anthocyanins in leaves and stems [37]. *TT4* along with *TRANSPARENT TESTA8* (*TT8*), a bHLH DNA-binding superfamily protein which is required for normal expression of *DFR* [38] associated with MYB113 in the interaction network. An important candidate TF for spider mite resistance is WD40 protein, which was expressed only during mite infestation in the resistant cultivar. Many studies have shown that a conserved MYB-bHLH-WD40 (MBW) regulatory complex control the expression of anthocyanin biosynthesis genes [39]. So, we deduced that MBW regulatory complex has the same

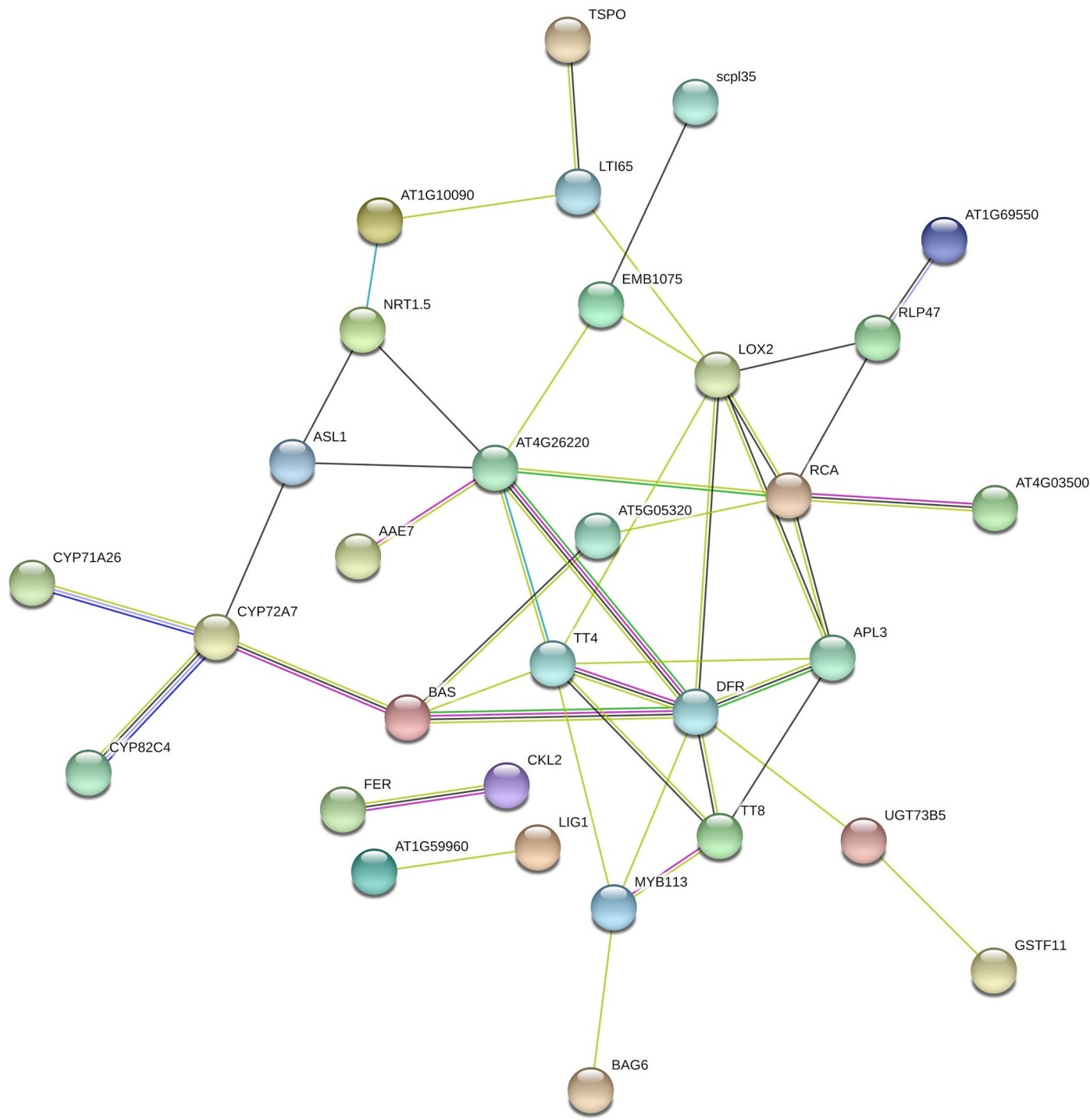

**Fig 4. Interaction networks of up-regulated genes identified in resistance cultivar as compared to susceptible cultivar at day 0 (control samples).**

function in common bean and is involved in resistance. In addition, another TF for spider mite resistance may be TCP protein, whose expression was increased during spider mite infestation in the resistant cultivar. Recent studies suggested that TCP proteins play an important role in systemic acquired resistance (SAR) which is induced plant immunity, activated by

pathogen infection [40–42]. Additionally, we identified a gene encoding WRKY50 which was significantly up-regulated in resistant cultivar during the middle-to-late stages (at day 5) of spider mite feeding. The spider mite infestation also affected the expression levels of Cys2/His2-type and C2H2-type zinc finger proteins as up-regulated genes in resistance and susceptible cultivars at day 5 post-infestation, respectively. The Cys2/His2-type zinc finger proteins are not only related to plant stress responses, but also enhance the resistance against pathogen infection [43]. ERFs, another important group of TFs, which play roles in integrating ET/JA signals [31, 44], activating the phenylpropanoid biosynthetic pathway and expression of resistance genes [45, 46], were activated in resistance cultivar at day 5 post-infestation. We observed increased transcript abundance of two key ERFs, including ERF1 and ERF9. The role of ERF1 as a regulator of ethylene responses after pathogen attack has been documented in *Arabidopsis* [47] but ERF9 has not proven to be relevant in the defense responses.

## Hormone regulation

GO enrichment analysis showed other GO terms that significantly overrepresented among up-regulated genes of resistant cultivar, including "response to stress", "response to stimulus" and "response to jasmonic acid". JA signaling has closely been associated with defense mechanisms against pathogens and insects [48, 49]. In our study, *13S-lipoxygenase 2* (*LOX2*), a JA signaling and biogenesis gene, was detected as up-regulated gene in resistant cultivar, where it had a low transcript abundance in susceptible cultivar. Up-regulation of *allene oxide synthase* (*AOS*), needed to JA production, at day 1 post-feeding suggests that common bean resistance to the disease is enhanced by the activation of JA signaling pathways. Our result is corroborated with the previous studies where the expression levels of *LOX* and *AOS* significantly increased after infestation [27, 50, 51]. In addition, the expression of two auxin signaling pathway genes, *SAUR* (small auxin-up RNA) and *ARF5* (Auxin response factor 5), were down- and up-regulated in resistant cultivar, respectively. *SAUR* genes are related to cell division[52, 53] and reportedly regulated by the auxin level, indicating that this process could be impaired by spider mite feeding. The down-regulation of *SAUR* gene is in agreement with the previous studies on *A. thaliana* [54] and soybean [55].

## Pathogen elicitor perception

In our study, only one disease resistance protein from TIR-NBS-LRRs class (Phvul.005G093 400.1) had higher transcript abundance in resistance cultivar, while one TIR-NBS-LRR (Phvul.010G029800.1), two NB-ARC domains-containing (Phvul.002G130666.1, Phvul.010 G064700.1), and one CC-NBS-LRR (Phvul.003G247601.1) showed a high level of expression in susceptible cultivar before infestation. During the first stage of mite infestation, five TIR-NBS-LRRs (Phvul.002G323100.1, Phvul.002G323400.2, Phvul.004G046400.1, Phvul.011G14 0300.1 and Phvul.010G132433.1), three NB-ARC (Phvul.003G002926.1, Phvul.004G013300.1 and Phvul.008G071300.3) and one leucine-rich repeat (LRR) protein kinase (Phvul.007G0875 50.1) were highly expressed in resistance cultivar, whereas one TIR-NBS-LRR (Phvul.004G058 700.1) and three NB-ARC (Phvul.008G031200.10, Phvul.010G064700.1, Phvul.011G195400.1) were found to be up-regulated in susceptible common bean cultivar. But these genes could not fully exert their expression with the extension of infestation time except one NB-ARC (Phvul.010G064700.1) in susceptible cultivar. At the first glance, it seems that the susceptible cultivar has a higher number of up-regulated disease resistance genes than the resistant one before mite infestation. But our results indicated that the response of the resistant plants was more robust than that of the susceptible cultivar upon pathogen attack. This can be elucidated by the role of miRNAs in down-regulating defense-related genes expression in susceptible

cultivar [56, 57]. The non-specific defense responses to deter the pathogen can also explain loss of defense-related genes expression at fifth day.

In addition, one gene encoding the Cysteine-rich RLK (CRK10) was highly expressed upon spider feeding. This highly up-regulated *CRK* gene seems to indicate its potential role in resistance against spider mite. We also observed receptor-like proteins (RLPs) and receptor-like kinase (RLK) genes that have a direct effect on the pathogen in the both cultivars [58].

## Antioxidant and detoxification processes

Reactive oxygen species (ROS) are involved in various processes along the plant life, but are best known as a key component of the signaling events involved in abiotic and biotic stress responses, so that are rapidly induced and accumulated after pathogen attack [51, 59]. An important response to control ROS is the induction of scavenging genes. In this respect, heat shock proteins (HSPs) play an important role in supporting ROS scavenging activity and stress tolerance [60]. In our study, HSP 70 was found to be up-regulated during spider mite feeding only in resistant cultivar, which highlight the function of HSPs in plant defense against pathogenic infection and reduce accumulation of ROS. This notion can be supported by the *P. vulgaris-Colletotrichum* interaction study [61], that HSPs are highly expressed against *Colletotrichum lindemuthianum* infection. Among detoxification genes, UDP-glycosyltransferase significantly up-regulated in resistant cultivar at all-time points, suggesting this gene may play an important role in the common bean resistance to spider mite feeding. The previous studies conducted on nematode attack in wheat [62] and *Fusarium* in *Brachypodium distachyon* [63] reinforces our argument about UDP-glycosyltransferase function.

## Cell wall

Xyloglucan endotransglucosylase/hydrolases (XTHs) are a family of enzymes that facilitate cell wall expansion [64] and also have the functions, probably associated with resistance mechanism [65]. In current study, we observed two XTHs (XTH22: Phvul.003G147700 and XTH9: Phvul.005G111300) in resistant cultivar significantly expressed at day 1 post infestation. Another important candidate gene for spider mite resistance may be the malectin-like receptor kinase FERONIA (FER), which was up-regulated and showed increase abundance during mite infestation, although there is no significant information on the role of this gene in response to feeding.

## Other genes

There are substantial reports regarding expression of pathogenesis-related (PR) genes under numerous stresses in common bean [61, 66]. Our transcriptome study successfully identified one PR-5 like receptor kinase (Phvul.004G155500) as up-regulated gene in resistant cultivar at day 5, which has been implicated in plant disease resistance, induced by different pathgens and share significant sequence similarity in many species [61, 66] 2-oxoglutarate (2OG) and Fe (II)-dependent oxygenase is another up-regulated gene in resistance cultivar under infestation which make it a potential candidate gene for resistance against spider mite and probably suitable for breeding programs. This gene has been previously described as responsive to pathogens [27].

## KEGG pathway functional enrichment analysis of the DEGs

Based on the KEGG pathway enrichment analysis, flavonoid biosynthesis, biosynthesis of secondary metabolites, metabolic pathways, phenylpropanoid biosynthesis and Linoleic acid

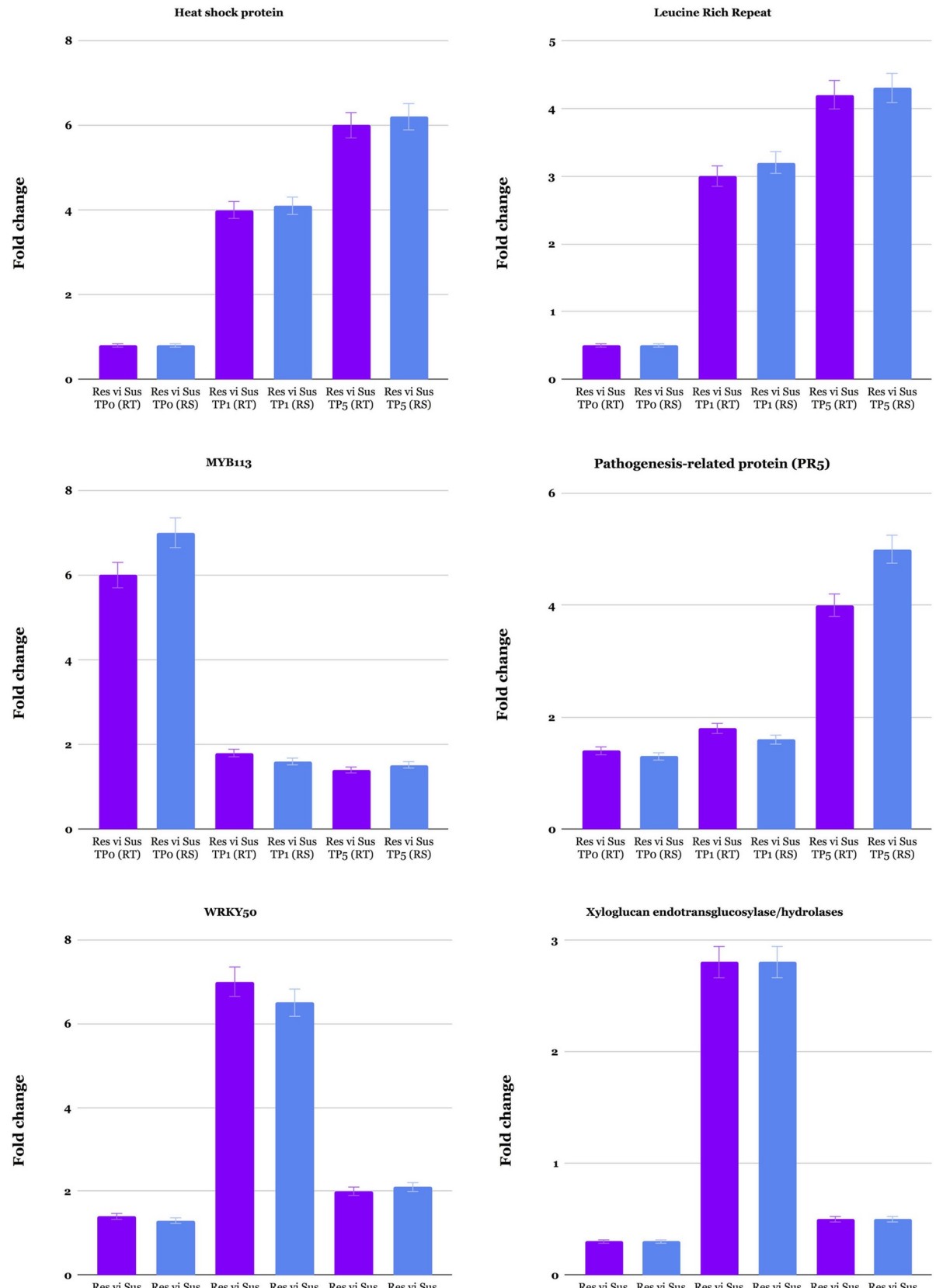

**Fig 5. qRT-PCR results of genes selected from the RNA-Seq analysis of common bean–spider mite interaction.** Expression levels of tested genes were normalized based on of *Actin* gene and then compared to relative expression values determined by RNA-Seq. Relative expression values of samples were determined by using the average expression value of all replicates of a particular group. Standard deviation among replicates is represented by error bars. Res and Sus represent resistant and susceptible cultivars. TP0, TP1 and TP5 represent first, second and third time points. RT ans RS in parentheses represent qRT-PCR and RNA-Seq.

metabolism were found to be the most changed pathways (S6–S8 Tables). In this assay it was demonstrated that the metabolic pathway biosynthesis of secondary metabolites contained the largest number of DEGs. From the results, we found that KEGG pathways, such as plant–pathogen interaction, plant hormone signaling transduction, and glucosinolate biosynthesis pathways play important roles in defence responses of common bean to spider mite.

## Validation of DEGs by using qRT-PCR

In order to verify gene expression results of transcriptome data analysis, six DEGs having annotations were selected for qRT-PCR analysis. They include the genes encoding pathogenesis-related proteins PR5, heat shock protein, leucine rich repeat, MYB113, XTH and a WRKY50 (Fig 5). Quantitative RT-PCR analysis was conducted on 12 RNA samples that were used in the preparation of sequencing libraries. Relative expression profiles of DEGs in the both resistant and susceptible evaluated using qRT-PCR were in complete agreement with the RNA-Seq data. This is in line with other studies, which showed almost the same level of fold changes between RNA-Seq data and qPCR [67, 68].

## Conclusion

To our knowledge, this investigation is the first study to identify molecular mechanisms involved in the common bean resistance to spider mite feeding by using RNA sequencing technology. In summary, DEGs were identified for control samples,1 and 5 days after infestation of spider mite in resistant and susceptible cultivars of common bean. Importantly, we identified secondary metabolism, multiple disease resistance proteins, TFs and genes involved in cell wall expansion and antioxidant processes that were modulated by spider mite attack. Overall, this study extended our understanding of the defense molecular mechanisms of two common bean cultivars with different genetic backgrounds during spider mite infestation. We came to the conclusion that these data provide important and valuable information for future research in common bean.

## Supporting information

**S1 Table. List of DEGs among time points in resistant cultivar.**
(XLSX)

**S2 Table. List of DEGs among time points in susceptible cultivar.**
(XLSX)

**S3 Table. List of DEGs between cultivars at first time point.**
(XLSX)

**S4 Table. List of DEGs between cultivars at second time point.**
(XLSX)

**S5 Table. List of DEGs between cultivars at third time point.**
(XLSX)

**S6 Table. Distribution of KEGG enriched pathways for DEGs between cultivars at first time point.**
(XLSX)

**S7 Table. Distribution of KEGG enriched pathways for DEGs between cultivars at second time point.**
(XLSX)

**S8 Table. Distribution of KEGG enriched pathways for DEGs between cultivars at third time point.**
(XLSX)

## Author Contributions

**Data curation:** Marie Shoorooei.

**Formal analysis:** Aboozar Soorni.

**Investigation:** Rahmat Mohammadi.

**Methodology:** Marie Shoorooei, Rahmat Mohammadi.

**Project administration:** Abdul Hadi Hoseinzadeh, Aboozar Soorni, Masoud Torkzadeh Mahani, Reza Maali Amiri, Hossein Allahyari.

**Resources:** Marie Shoorooei.

**Validation:** Aboozar Soorni.

**Visualization:** Aboozar Soorni.

**Writing – original draft:** Aboozar Soorni.

**Writing – review & editing:** Aboozar Soorni.

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
