## [Decision Letter · Decision Letter 0]

2 Dec 2019

PONE-D-19-29263

Comparative transcriptome provides molecular insight into defense-associated mechanisms against spider mite in resistant and susceptible common bean cultivars

PLOS ONE

Dear Dr Soorni,

Thank you for submitting your manuscript to PLOS ONE. After careful consideration, we feel that it has merit but does not fully meet PLOS ONE’s publication criteria as it currently stands. Therefore, we invite you to submit a revised version of the manuscript that addresses the points raised during the review process.

We would appreciate receiving your revised manuscript by Jan 16 2020 11:59PM. To enhance the reproducibility of your results, we recommend that if applicable you deposit your laboratory protocols in protocols.io, where a protocol can be assigned its own identifier (DOI) such that it can be cited independently in the future. For instructions see: http://journals.plos.org/plosone/s/submission-guidelines#loc-laboratory-protocols

We look forward to receiving your revised manuscript.

Kind regards,

Kandasamy Ulaganathan

Academic Editor

PLOS ONE

Journal Requirements:

Additional Editor Comments (if provided):

Completely revise the manuscript by taking into considerations all the points raised by the reviewers and incorporate all corrections pointed out by the reviewers.

Reviewers' comments:

Reviewer's Responses to Questions

**Comments to the Author**

1. Is the manuscript technically sound, and do the data support the conclusions?

Reviewer #1: Partly

Reviewer #2: Yes

Reviewer #3: Yes

2. Has the statistical analysis been performed appropriately and rigorously? 

Reviewer #1: No

Reviewer #2: Yes

Reviewer #3: Yes

3. Have the authors made all data underlying the findings in their manuscript fully available?

Reviewer #1: No

Reviewer #2: Yes

Reviewer #3: Yes

4. Is the manuscript presented in an intelligible fashion and written in standard English?

Reviewer #1: No

Reviewer #2: Yes

Reviewer #3: Yes

5. Review Comments to the Author

Reviewer #1: It’s known that the two-spotted spider mite Tetranychus urticae Koch is one of the economically most important pests in a wide range of outdoor and protected crops worldwide. To identify the resistant mechanism against spider mite of host plant will greatly improve the crops breeding. Authors used RNA-seq to investigate the candidate resistant related genes that’s a good idea to get the whole deferential expression genes (DEGs) between resistant and susceptible cultivars. However this manuscript lacks a systematic statement, reader hardly to read and to get the useful information. Authors may include some articles which addressed on the study on resistant mechanism of spider mite to explain the relationship of DEGs and resistant mechanism. (such as: (1) Can Plant Defence Mechanisms Provide New Approaches for the Sustainable Control of the Two-Spotted Spider Mite Tetranychus urticae?. Int J Mol Sci. 2018;19(2):614. (2) Acaricide resistance mechanisms in the two-spotted spider mite Tetranychus urticae and other important Acari: a review. Insect Biochem Mol Biol. 2010 Aug;40(8):563-72.)

For major part:

1. P7, Line 167: The data was not consistent to Table S3. In the manuscript the DFEs are 48 for up-regulation and 46 for down-regulation, while only 93 DFEs were included in Table S3.

2. From P8, Line 204 to P12, Line 318: There was no related data to show the expression level of DFEs. Such as “The two most induced genes involved in flavonoid….” (Line 204), “CHS showed more than 7-fold change increase in resistant cultivar. O-methyltransferases (OMTs), involved in phenylpropanoids, flavonoids, and anthocyanin methylation, was up-regulated in resistant cultivar and showed an 8-fold increase in expression level upon spider mite feeding.” and so on. Authors need to construct some tables to indicate the expression profiles of DGEs.

3. The results of Figure 5 were not consistent to Table S4 and Table S5. In Figure 5 the fold change of heat shock protein@TP1, Leucine Rich Protein@TP1, PR5@TP5, WRKY50@TP1 were significantly up-regulated in resistant cultivar (fold change >4) in RNA-seq analysis. The Table S3 and S5 collected the DFEs with Log2 FPKM (fold change) ≥1.0. However, they did not present in the Table S4 (@TP1) or Table S5 (@TP5).

4. P4, Line 101, Authors need to explain why they set the time point as 0, 1, and 5 day after infection?

5. The Figure 2 presents the comparison of Up_Res_T0T1 vs. Up_Sus_T0T5, and Down_Res_T0T1 vs. Down_Sus_T0T5. There are two variants: cultivar and treatment time. That cause the comparison none senesce.

6. P12, Line 332: Please explain why authors select these 6 genes for validation, especially some were not included in the previous parts of Result and Discussion.

7. There is rare discussion on the defense-associated mechanisms against spider mite in resistant and susceptible common bean cultivars.

The Minor Part:

1. P2, Line 41: It should be “spider mite”.

2. P3, Line 73: TSSM is not the abbreviation of “Tetranychus urticae Koch”, put the correct one.

3. P4, Line 100: What does “In six-leaf stage of Meier” mean?

4. P5, Table 5: Since there are many Gene ID related to same functional annotation, please clarify, such as “Pathogenesis-related protein, PR5” for the first item.

5. P6. Line 147: This part is not important. Please move it to supplemental information.

6. The full name should be showed at its first present, such as CYP at P8, Line 215.

7. P8, Line 216: Authors mentioned “In the susceptible genotype, CYP83B1 gene required for the synthesis of indole glucosinolates, was down regulated during infestation which is not consistent with previous reports in a number of researches”. Please explain your opinion.

8. P12, Line 312: “The previous studies conducted on nematode attack in wheat (Qiao et al, 2018) and Fusarium in Brachypodium distachyon (Schweiger et al, 2013) reinforces our argument about UDP-glycosyltransferase function.” These two references were cited in a wrong format and were not included in reference list.

Reviewer #2: I think this can be an informative manuscript after adding more data to it. I added some of my recommendations as comments in the manuscript and have some more recommendations as follows: first of all, you need to perform pathway analysis and add the results to the manuscript, I recommend using KOBAS 3.0. The second thing, I already added comments on Venn diagrams, just wanted to emphasis it more here, you want to follow that and revise the manuscript accordingly. I don't see a table or graph showing the number of DEGs that you find from the comparisons that you made, you need to add it to the manuscript. I don't know if it is problem with the system that I don't see all supportive information, or something else, you do have referred to tables such as "S3-5" in the manuscript that I cannot find them. The ones that I have access to them, Table S1-S5, I don't see any column labels and title, these tables are really confusing, you need to fix it. You also want to check all gene names in the manuscript, the all need to be italicized. Your figures text is not very clear, especially Fig 5, you need to revise all of the figures and be sure that quality matches journal requirements.

Reviewer #3: The authors compared the transcriptome of resistant and susceptible common bean cultivars after infection of spider mite, aiming at shed light into the understanding of mechanisms and biological pathways of resistance. Materials and methods are not sufficiently detailed to understand the experimental design. English grammar, as it stands, needs to be improved; and some sentences are wordy and need to be shorted. Authors need to answer to the queries below, in order to resubmit to PLOS ONE for a final round and acceptance. For detailed comments, see below:

Page 4:

1.Lines 96-97: Information of specimen collection and rearing method of insect should be described in more detail.

2.Lines 97-100: Plant growth information of two cultivars should be described in more detail.

3.Line 100: In six-leaf stage of Meier [13], is this phrase right? I noticed that the “Meier” is a surname.

4.Line 100: Why only the female mites were selected? Whether these female mites were mated or not?

5.Line 100: The common bean cultivars growing to “six-leaf stage”, the common bean cultivars need to take how many days to grow to this stage? And I think this would be added in the text.

6.Line 101: As spider mite is very small, how many individuals were used as each replicate?

7.Line 101: Why the authors set three time points （0, 1and 5 days）? As I known, in some studies about defense-associated mechanisms against insect infestation, more time points (0, 12, 24, 48, and 72 h or more hours) are set after insects’ infection.

8.Lines 100-101: What the meaning of “the same-aged”? Did the mites possess the same time span of ecdysis, forexample, 24 hours after ecdysis of these mites?

9.Lines 100-101: The sentences “45 same-aged adult female mites were placed on sixth leaf of cultivars.” Should be described in more detail. Because I was not clear how many inviduduals placed on each leaf of each cultivar per replicate.

10.Lines 100-101: This sentence is too long, and the grammar needs to be improved.

11.Lines 105: The authors mentioned in line 98 “The experiment was conducted

using a factorial experiment based on completely randomized design with three replicates”, however in line 105, they mentioned that “Total RNA was extracted from two biological replicates”. Generally, three replicates are taken for each cultivar.

Page 5:

1.Lines 134: the authors only six DEGs for validating candidate DEGs, I suggest that more related DEGs are need to validate.

2.Lines 141: The 2-ΔΔCt should correct to be 2-ΔΔCT.

6. PLOS authors have the option to publish the peer review history of their article (what does this mean?). If published, this will include your full peer review and any attached files.

Reviewer #1: No

Reviewer #2: No

Reviewer #3: No

---

## [Author Response · Author response to Decision Letter 0]

5 Jan 2020

First of all, I would like to thank your editorial work over our manuscript. We also would like to thank the reviewers for their work and for some of the suggestions that they made. We think that the incorporation of these suggestions have increased the quality of the manuscript. We are pleased with the revision. We have addressed the concerns presented by the reviewers in the new version of the manuscript. We also have answered their questions and clarify their points in rebuttal letter. If you have any new comment or question, please do not hesitate to contact me.

---

## [Decision Letter · Decision Letter 1]

22 Jan 2020

Comparative transcriptome provides molecular insight into defense-associated mechanisms against spider mite in resistant and susceptible common bean cultivars

PONE-D-19-29263R1

Dear Dr. Soorni,

We are pleased to inform you that your manuscript has been judged scientifically suitable for publication and will be formally accepted for publication once it complies with all outstanding technical requirements.

With kind regards,

Kandasamy Ulaganathan

Academic Editor

PLOS ONE

Additional Editor Comments (optional):

Reviewers' comments:

Reviewer's Responses to Questions

**Comments to the Author**

1. If the authors have adequately addressed your comments raised in a previous round of review and you feel that this manuscript is now acceptable for publication, you may indicate that here to bypass the “Comments to the Author” section, enter your conflict of interest statement in the “Confidential to Editor” section, and submit your "Accept" recommendation.

Reviewer #1: All comments have been addressed

Reviewer #2: All comments have been addressed

2. Is the manuscript technically sound, and do the data support the conclusions?

Reviewer #1: Partly

Reviewer #2: Yes

3. Has the statistical analysis been performed appropriately and rigorously? 

Reviewer #1: Yes

Reviewer #2: (No Response)

4. Have the authors made all data underlying the findings in their manuscript fully available?

Reviewer #1: Yes

Reviewer #2: (No Response)

5. Is the manuscript presented in an intelligible fashion and written in standard English?

Reviewer #1: Yes

Reviewer #2: Yes

6. Review Comments to the Author

Reviewer #1: Your manuscript entitled "Comparative transcriptome provides molecular insight into defense-associated

mechanisms against spider mite in resistant and susceptible common bean cultivars" was accepted.

Reviewer #2: (No Response)

7. PLOS authors have the option to publish the peer review history of their article (what does this mean?). If published, this will include your full peer review and any attached files.

Reviewer #1: No

Reviewer #2: No

---

## [Editor Report · Acceptance letter]

27 Jan 2020

PONE-D-19-29263R1 

Comparative transcriptome provides molecular insight into defense-associated mechanisms against spider mite in resistant and susceptible common bean cultivars 

Dear Dr. Soorni:

I am pleased to inform you that your manuscript has been deemed suitable for publication in PLOS ONE. Congratulations! Your manuscript is now with our production department. 

With kind regards,

on behalf of

Dr. Kandasamy Ulaganathan 

Academic Editor

PLOS ONE